# Self-Supervised Learning Methods for Label-Efficient Dental Caries Classification

**DOI:** 10.3390/diagnostics12051237

**Published:** 2022-05-16

**Authors:** Aiham Taleb, Csaba Rohrer, Benjamin Bergner, Guilherme De Leon, Jonas Almeida Rodrigues, Falk Schwendicke, Christoph Lippert, Joachim Krois

**Affiliations:** 1Digital Health & Machine Learning, Hasso Plattner Institute, University of Potsdam, 14469 Potsdam, Germany; benjamin.bergner@hpi.de (B.B.); christoph.lippert@hpi.de (C.L.); 2Department of Oral Diagnostics, Digital Health and Health Services Research, Charité—Universitätsmedizin Berlin, 10117 Berlin, Germany; csaba.rohrer@charite.de (C.R.); falk.schwendicke@charite.de (F.S.); joachim.krois@charite.de (J.K.); 3Contraste Radiologia Odontológica, Blumenau 89010-050, SC, Brazil; guilherme@contrasteradiologia.com; 4Department of Surgery and Orthopedics, School of Dentistry, Universidade Federal do Rio Grande do Sul—UFRGS, Porto Alegre 90010-460, RS, Brazil; jorodrigues@ufrgs.br; 5Hasso Plattner Institute for Digital Health at Mount Sinai, Icahn School of Medicine at Mount Sinai, New York, NY 10029, USA

**Keywords:** unsupervised methods, self-supervised learning, representation learning, dental caries classification, data driven approaches, annotation efficient deep learning

## Abstract

High annotation costs are a substantial bottleneck in applying deep learning architectures to clinically relevant use cases, substantiating the need for algorithms to learn from unlabeled data. In this work, we propose employing self-supervised methods. To that end, we trained with three self-supervised algorithms on a large corpus of unlabeled dental images, which contained 38K bitewing radiographs (BWRs). We then applied the learned neural network representations on tooth-level dental caries classification, for which we utilized labels extracted from electronic health records (EHRs). Finally, a holdout test-set was established, which consisted of 343 BWRs and was annotated by three dental professionals and approved by a senior dentist. This test-set was used to evaluate the fine-tuned caries classification models. Our experimental results demonstrate the obtained gains by pretraining models using self-supervised algorithms. These include improved caries classification performance (6 p.p. increase in sensitivity) and, most importantly, improved label-efficiency. In other words, the resulting models can be fine-tuned using few labels (annotations). Our results show that using as few as 18 annotations can produce ≥45% sensitivity, which is comparable to human-level diagnostic performance. This study shows that self-supervision can provide gains in medical image analysis, particularly when obtaining labels is costly and expensive.

## 1. Introduction

Medical imaging plays a vital role in patient and dental healthcare. It aids in disease prevention, early detection, diagnosis, and treatment. However, efforts to employ machine learning algorithms to support in clinical settings are often hampered by the high costs of required expert annotations. Generating expert annotations of dental data at scale is non-trivial, expensive, time-consuming, and is associated with risks on privacy leakages. Even semi-automatic software tools may fail to sufficiently reduce annotation expenses [1]. Consequently, the scarcity of data annotations is one of the main impediments for machine learning applications in (dental) healthcare. At the same time, unprecedented amounts of *unlabeled multimodal* data are collected in clinical routines, calling for solutions to exploit such rich data [2]. In dentistry, so far, almost all machine learning applications followed the supervised learning paradigm, e.g., [3]. The amount of labeled data used varies between the subfields of dentistry but also among types of application. For instance, Kim et al. [4] used 12,179 labeled panoramic radiographs to detect periodontal bone loss, whereas Setzer et al. [5] used only 20 CBCT volumes to detect periapical lesions. Notably, both studies reported diagnostic performances that are considered to be clinically useful. Given, however, that an estimated global annual total of about 520 million dental x-ray examinations are conducted [6], the referenced datasets are only scratching the surface, with respect of being comprehensive or representative. This in turn hinders transferabilty and generalizability, and finally the dissemination of machine learning applications into the clinical setting.

In contrast to supervised learning approaches, self-supervised representation learning provides a viable solution when labeled training data is scarce. In general, unsupervised representation learning aims to construct an embedding space, in which data samples that are semantically similar are closer to each other, and those that are different are farther apart. The self-supervised family learns this representation space by creating a supervised (proxy) task from the data itself. In other words, the supervisory signals are derived from the data. The resulting semantic representations are stored in the form of deep neural network weights. Subsequently, the representations (embeddings) that solve the proxy task will also be useful for other real-world downstream tasks, e.g., image classification or segmentation, hence reducing the burden of manual annotation. This two-phase learning scheme in self-supervised learning, illustrated in Figure 1, is similar to Transfer Learning, except that here the pretraining phase requires no human labels. Self-supervised methods differ in the created (proxy) task used to learn representations from unlabelled data.

Self-supervised learning has found many application fields [7], and has shown to improve the data and annotation efficiency of deep learning models. In the medical imaging domain, these methods have witnessed a recent surge of interest [8,9]. Early works of self-supervision in the medical imaging context targeted specific use-cases such as depth estimation in monocular endoscopy [10], medical image registration [11], body part recognition [12], in-disc degeneration using spinal MRIs [13], body part regression for slice ordering [14], and several others [15,16,17,18]. Many of these works made assumptions about input data, resulting in engineered solutions that have limited generalization to other downstream tasks. Therefore, several works proposed to employ proxy (pretext) tasks for self-supervision from medical scans. For instance, Tajbakhsh et al. [19] used orientation prediction from medical images, Spitzer et al. [18] predicted the 3D distance between two 2D patches in brain scans, Jiao et al. [20] relied on the order of ultrasound video clips and random geometric transformations, Zhou et al. [21] extended image reconstruction techniques to 3D medical scans, and Taleb et al. [22] extended five self-supervised methods to the 3D medical context. Many other works created proxy tasks for the medical imaging domain [23,24,25,26]. Recently, contrastive learning methods have been applied to medical scans [22,27,28,29], where they showed promising results on several medical image analysis tasks. Similar to proxy task learning, contrastive methods learn general purpose semantic representations from unlabeled data. However, the underlying mechanism of these methods is that they aim to learn representations that are invariant under different data augmentations. This is achieved by maximizing similarity of representations obtained from different augmented versions of each sample. Our work follows this line of algorithms, and we employ contrastive methods on dental data. To the best of our knowledge, this work is the first to evaluate the self-supervised learning scheme in the dentistry domain.

In this study, we evaluate self-supervised deep learning algorithms on dental caries classification. Dental caries is the most prevalent health disease, affecting more than 3 billion people worldwide [30]. For diagnosis, clinicians commonly analyze bitewing radiographs (BWRs). Notably, the assessment of BWRs by dentists is associated with low sensitivity and shows considerable interexaminer variation [31,32]. The growing quantities of dental data and the challenging nature of dental caries detection motivated employing deep learning techniques for this task [4,33,34,35]. Additionally, the high costs associated with labeling caries in bitewing radiographs (BWRs), make this domain a pertinent test-bed for self-supervised representation learning algorithms. For instance, annotating the curated test set for this study (see Section 2.1) required 71 man-hours approximately. At the same rate, annotating the full training dataset would have required more than 7600 man-hours (∼950 work-days). In this work, we aim to test the following hypotheses:Does pretraining of models with self-supervision improve the diagnostic performance of the model?Does self-supervision reduce the amounts of annotations required, i.e., improves label-efficiency?Does oversampling of the positive class (here, caries are present) improve the diagnostic performance of the classifier?

In the following sections, we first detail the dataset we used for both self-supervsed training and dental caries classification. Then, we provide the details of the employed self-supervised algorithms and related implementation details. Afterwards, we present the evaluation results of these algorithms on dental caries classification. Finally, we discuss the obtained results and highlight the gains of applying self-supervision in the dental domain.

## 2. Materials and Methods

### 2.1. Dataset

The dataset was collected by three dental clinics in Brazil, which are specialized in radiographic and tomographic examinations. The dataset consisted of 38,094 BWRs taken between 2018 and 2021. In total, 9779 patients with an average [min–max, sd] age of 34 [3–88, 14] years constituted the sample. The average [min–max, sd] number of scans per patient was 4 [1–11, 1]. We preprocessed the radiographs by extracting individual tooth images using a helper model, a deep-learning based tooth instance-segmentation model (unpublished). Each detected tooth was then cropped from the BWR using a bounding box that fully contained the tooth. The procedure resulted in a dataset of 315,786 cropped tooth images. Out of those, we observed 49.9% of molars, 40.5% of premolars and 9.6% of canines and incisors, respectively. It is noteworthy that the tooth classification with the helper model may not be perfect, but as we are interested in the tooth as an object, we ignore these imperfections in automated tooth labeling. Tooth-level caries labels were extracted from electronic health records (EHRs) that summarize the patient’s dental status. The dataset has a caries prevalence of 19.8%. Although EHR-based ground truth labels are known to come with uncertainties and biases [36], we found that they provide sufficiently rich signals (semantically) when fine-tuning self-supervised models. For model evaluation purposes, a hold-out test set was curated by dental professionals. The test set consisted of a random sample of 343 BWRs. The average [min-max, sd] age of the patients within the test set was 33 [5–80, 13]. The BWR samples were annotated for dental caries by 4 independent dentists. These annotations were reviewed by a senior dentist (+13 years of experience) to resolve conflicts and establish the ground truth in the test set. After extracting tooth-level images with the helper model, the test set contained 2846 tooth samples with 29.9% caries prevalence (850 positive and 1996 negative). We observed 49.2% molars, 40.5% premolars, and 10.3% canines and incisors, respectively. We ensured that the set of patients is independent in the training and the test datasets.

### 2.2. Self-Supervised Learning Algorithms

The basic idea of self-supervised learning is, that convolutional neural networks (CNNs) are trained to learn semantic data representations from *unlabeled* images, requiring no human labels. It is noteworthy that in the present study this pretraining stage is performed on the raw BWRs, even without tooth cropping using the helper model. Each algorithm results in a CNN encoder model that can be fine-tuned on subsequent downstream tasks, e.g., dental caries classification. Three algorithms were employed that all rely on introducing invariance to augmentations of input samples to the learned representations and recently excelled on natural imaging benchmarks [37,38,39]. These approaches build upon the cross-view prediction framework introduced in [40], e.g., predicting random crops of the same image from each other. Such approaches solve the problem in the feature space, i.e., the representation of an image view should be predictive of another view. However, predicting in feature space directly can lead to collapsed representations, i.e., a trivial constant solution across views. The chosen algorithms differ in their techniques to avoid such collapsed representations.

#### 2.2.1. SimCLR

First proposed by Chen et al. [37], this method follows the Contrastive family of algorithms [41,42], which rely on the Noise Contrastive Estimation (NCE) loss [43]. This loss aims to maximize the mutual information between related signals, in contrast to other signals, in the embedding space. SimCLR circumvents the aforementioned collapsed representations problem by reformulating the prediction problem into one of classification. To achieve that, SimCLR discriminates (classifies) artificially created “positives” and “negatives” from unlabeled data points, as illustrated in Figure 2a. The terms “positive” and “negative” in this context have no relation to labels whatsoever; here, they indicate views of the same image (positives) and of other images (negatives). Hence, we call them “same-views” and “others-views” to remove this confusion. This algorithm’s steps are:The image dataset is processed in batches, where same-views and others-views are created from each batch.For each input image, a pair of same-views is created using image augmentations. The others-views are then the remaining images in the batch.All images are then processed by the encoder network, to produce a vector representation for each image. We employ CNNs for encoder architecture, but other architectures are possible. During training, the encoder is replicated to process *pairs* of samples, constituting a Siamese architecture [44].Each representation is then processed by a small projection head, which is a non-linear multi-layer perceptron (MLP) with one hidden layer.Finally, the NCE loss computes the cosine similarity across all samples. This loss encourages the similarity between same-views to grow larger (attracts their representation vectors in the embedding space), and the similarity to others-views to become smaller (repels their representations in the embedding space).

#### 2.2.2. BYOL

This method was proposed by Grill et al. [38]. It attempts to avoid the creation of others-views mechanism used in SimCLR. The motivations are two folds. First, it can be computationally expensive, as the NCE loss may require a large number of others-views to learn rich representations. SimCLR [37] addresses this by using larger batch sizes (≥2048). Second, the semantics of others-views may require special treatment [45] to ensure they encourage better representations. Therefore, BYOL introduces asymmetric parameter updates to the encoder architecture as an alternative for the others-views generation mechanism. In other words, the two encoder models in the Siamese architecture, illustrated in Figure 2b, do not have identical weights. The process of how they are trained is explained below.

The Siamese architecture processes a pair of augmented views of each image, similarly to SimCLR. However, the architecture in BYOL is modified to be asymmetric by:The first *online* network is trained to predict the representations of the other *target* network.The weights of the target network are an exponential moving average of the online network.This means that the actual parameter updates, i.e., gradients of the loss, are applied on the *online* network only. This is ensured by a “stop gradient” technique on the target network, which has been found, empirically, to be essential [46] to avoid collapsed representations.The training loss is the mean squared error (MSE) between the predictions of online and target networks. Note that both networks use a projection head similar to SimCLR’s.

After training, only the encoder of the online network is kept, and everything else is discarded.

#### 2.2.3. Barlow Twins

This method, first proposed by Zbontar et al. [39], illustrated in Figure 2c, avoids both the others-views sampling of SimCLR and the asymmetric updates of BYOL by relying on a statistical principle called *redundancy reduction*. Its steps are:Assuming a batch of images. Two sets of augmented views are created by different augmentations.These views are processed concurrently with a Siamese encoder. Similar to SimCLR, the encoder weights are replicated, and the representations are projected with a projection head.The cross-correlation matrix of the two sets of representations is computed. Each entry of this matrix encodes the correlation between the corresponding representation entries.Finally, the loss is defined as the difference between the cross-correlation and the identity matrices. The intuition behind this is that it encourages the representations of same image views to be similar, while minimizing the redundancy between their components.

### 2.3. Image Augmentations in Self-Supervised Training

When training with the self-supervised algorithms, image augmentations are employed to create the same-view samples (also others-view samples in the case of SimCLR). The choice of augmentations influences downstream performance, as shown in [37,38,39]. In this study, we found that using the default augmentations by these methods proved unsuccessful (it did not improve caries classification results). We believe the nature of the data may have a role in this, i.e., medical images exhibit a more uniform nature than natural images, e.g., color distributions. Hence, we employed different image augmentations:Random resized cropping between 50–100% of input size.Random horizontal flip with 50% probability.Color adjustments (probabilities): Brightness (20%) and Contrast (10%), and Saturation (10%).Random rotation angles between −20${}^{\circ }$ to 20${}^{\circ }$.

We found the reduced probabilities of color adjustments to benefit the learned representations the most in our evaluations.

### 2.4. Implementation Details

All images were resized to the resolution of 384×384 pixels. We used the Resnet-18 [47] architecture as the neural network encoder. During self-supervision stage only, the used projection head has an output dimension of 128. For all training procedures, we employed the Adam optimizer [48]. During the self-supervised pretraining stage we trained with batch sizes of 224 images and set the initial learning rate to 0.001, while using cosine annealing [49]. After the self-supervised pretraining stage, the resulting encoder was employed in supervised dental caries classification, as illustrated in Figure 2d. To that end, a fully-connected layer with output units equal to the number of classes was added on top. In this stage we trained with a batch size of 92 images, set a fixed learning rate of 0.0001 and used the cross-entropy loss to learn from the EHRs labels. We did not tune the classification threshold and used a confidence score of 0.5 to discriminate between the positive and the negative prediction label. As evaluation metrics we computed ROC-AUC, sensitivity, and specificity. Our implementations were done in Python, using the libraries PyTorch v1.10.0, Pytorch-Lightning v1.5.4, and Lightly [50]. We ensured reproducibility of results by setting a unified random seed of 42 for all scripts and workers.

## 3. Results

In this section we report on model performance metrics of the supervised caries classification on tooth segments. Notably, for training we used EHR labels but all of the reported metrics are computed on the curated test set of 343 BWRs. According to the hypotheses defined above, we evaluated (1) whether fine-tuning based on pretrained models via self-supervision improved the diagnostic performance of the classifier compared to a baseline model that was initialized with random model weights, (2) if self-supervised pretraining improved the label-efficiency by successively adding more and more data to the model, and (3), if controlling the prevalence of the positive class (here, caries is present) in the training dataset improved the diagnostic performance of the classifier. We simulated that by successively oversampling tooth segments that showed caries lesions.

### 3.1. Fine-Tuning on Pretrained Models

In this set of results, we present the performance of models that were fine-tuned (trained) on the full image dataset (38,094 BWRs and 315,786 tooth segments) using the EHR labels. The models are initialized at the beginning of the fine-tuning phase, with sets of model weights that were obtained by the methods SimCLR, BYOL and Barlow Twins, respectively. As a baseline, we compare those to a model that was trained from scratch, i.e., whose weights are initialized randomly. The evaluation results for this set of experiments are shown in Table 1.

The highest sensitivity, with 57.9% was observed for Barlow Twins, followed by SimCLR and BYOL, with 57.2% and 54.6%, respectively. These values are considerably higher than 51.8%, obtained by the baseline model. The difference between the best performing model, here Barlow Twins, and the baseline is ∼6%. For specificity all models perform similarly, with the baseline model and the BYOL method showing slightly higher values (both at 91.3%), compared to SimCLR (89.3%) and Barlow Twins (88.9%). With respect to the ROC-AUC values, all pretrained models are close (73.3%, 73% and 73.4% for SimCLR, BYOL and Barlow Twins, respectively) but consistently higher than the baseline model (71.5%).

### 3.2. Data-Efficiency by Successively Increasing the Size of the Training Set

The results in this section demonstrate the obtained gains in data efficiency. For that purpose, we report on the performance of caries classification for different dataset sizes (up to 10% of the total dataset, which corresponds to ∼3.8K BWRs or 30K tooth segments) that were used for training (fine-tuning). Fine-tuning for all experiments was done for a fixed number of epochs (50 epochs each). For each subset, we compared the performance of the fine-tuned models to the baseline that was trained from scratch (random weight initialization). We repeated this process for each subset for five times to account for random sampling effects. The samples at each iteration were chosen randomly, resulting in ∼20% caries prevalence, which was close to the actual prevalence of the full dataset of 19.8%.

As shown in Figure 3 and Table 2 (see first 6 rows that are grouped by the prevalence of ∼20%) there was a slight tendency, but not a monotonic increase, towards higher scores for all model initialization methods. For SimCLR, BYOL, Barlow Twins and the baseline model sensitivity values ranged from 40.02% to 54.8%, from 44.78% to 55.32%, from 41.01% to 51.34%, and from 32.87% to 50.19%, respectively. The baseline model was performing worse compared to the pretrained models over all dataset sizes. Interestingly, the model pretrained with the Barlow Twins method obtained a sensitivity value of 46.28% even when fine-tuned with only 18 BWRs. With respect to specificity, SimCLR, BYOL, Barlow Twins and the baseline model values ranged from 55.39% to 79.42%, from 51.61% to 83.98%, from 58.43% to 86.85%, and from 63.30% to 81.57%, respectively. Here, in contrasts to above, the baseline model performed equally well (sometimes even better), compared to the pretrained models. It appeared that learning the representations for negative, non-caries tooth segments is easier, compared to learning positive representations of caries. The ROC-AUC values for all model configurations were alike and showed values in the ranges from 52% to ∼69%. However, for low data regimes (≤3K images) the ROC-AUC values were higher for Barlow Twins and BYOL compared to SimCLR and the baseline model in the balanced case (50% prevalence). Barlow Twins, in particular, exhibited improved values for this metric in most settings.

### 3.3. Oversampling of the Positive Class

In another experiment, we repeated the procedure as detailed above, however, we purposely oversampled tooth segments from the caries (positive) class and, hence, artificially increased the the prevalence of caries in the different training sets. We oversampled the caries class to 50% of input data (balanced case), and to 75%. The evaluation results are shown grouped by the prevalence of caries in the training dataset in Table 2 and Figure 3.

Similar to the first series of experiments, pretrained models outperformed the baseline model with respect to sensitivity. With increasing prevalence in the training set the sensitivity scores increased, from 40–50%, to 50–60%, to 79–80% for a prevalence of ∼20%, 50% and 75% in the training dataset, respectively. In the case of specificity, the differences between the models were negligible. However, with increasing prevalence the specificity scores tended to decrease, hence pointing out the trade-off between the frequency of positive and negative classes to be learnt from, and the model’s capacity of prediction making. However, it was worth noting that scores for ROC-AUC, which is considered a more balanced metric as it takes both sensitivity and specificity into account, did not change under different regimes of caries prevalence in the training dataset. ROC-AUC ranged from >50% to <70% and increased due to the dataset size but not in relation to the caries prevalence in the training set.

## 4. Discussion

In this present study we leveraged self-supervision to pretrain models on a large dataset stemming from routine care. We further, assessed the effect of self-supervised pretraining on a real-world supervised learning task, by training caries prediction models on EHR data and evaluating them on a test set with revised ground-truth labels.

We showed that for the downstream task of caries prediction pretraining with self-supervised algorithms provided a considerable performance boost, especially for the sensitivity of the model. All three presented methods outperformed the baseline, and added up to 6% in sensitivity (see Table 1). Hence, we accept the hypothesis that pretraining of models with self-supervision improves the diagnostic performance of a classifier. The effectiveness of leveraging pretrained models for boosting downstream prediction tasks, often referred to as transfer learning, is a well known and widely popular practice to boost the performance of machine learning models [51,52]. In this study, however, we showed that pretraining can be effectively done on domain-specific image data via self-supervision and does not have to stem from large open (non-medical) image databases such as ImageNet [53] or others [54,55]. It remains beyond the scope of the present study to evaluate the differences between different approaches of model pretraining.

This study also showed that pretrained models outperform the baseline with a significant margin when using few training samples, particularly in terms of sensitivity. In a low-data regime only as few as 18 BWR samples (=152 tooth images) yielded ≥ 45% sensitivity, which is competitive compared to the diagnostic performance of domain experts, who reportedly show sensitivities of around 47% (95% confidence interval (CI) 40% to 53%) [32]. This behavior was consistent even when the prevalence of positive labels was larger. In fact, the sensitivity margin was maximized (∼22%) between Barlow-Twins and the baseline, when using 75% positive labels (see Table 2). Hence, by using self-supervision techniques, the annotation process and data-efficiency may be improved as only a fraction of labeled data is required to achieve competitive results. We therefore accept the hypothesis that self-supervision may reduce the amounts of annotations required, i.e., improves data-efficiency.

Lastly, we shed light on the question if the prevalence of caries in the training set impacts the classifiers performance. Therefore, we ran different experiments while oversampling the positive class which yielded training sets with caries prevalences of ∼20%, 50%, and 75%, respectively. We observed an increase in scores for the sensitivity but at the expense of specificity scores. This is to be expected, as the number of positive samples (the minority class) grows larger, and hence the model did learn a richer representation of the positive class. However, when monitoring the AUC-ROC metric, which balances sensitivity and specificity, the prevalence in the training dataset did not impact the diagnostic performance of the classifier. Hence, we reject the hypothesis that oversampling of the positive class improves the diagnostic performance of the classifier. Though, another implication of this result was that for the purpose of auditing, reviewing, or reporting on the diagnostic performance of machine learning models, researchers should provide information about the prevalence of the positive class in the training dataset, so that the comparisons among different studies can be drawn more robustly.

The study comes with a number of strengths and weaknesses that need to be discussed in more detail. First, the study is, to our best knowledge, the first to showcase the potential of self-supervision in field of dentistry. This exciting technology may be well suited to address the hundreds of millions of new X-Ray images that are generated each year across the globe. Only if we can learn how to capture the existing variability and heterogeneity may we minimize biases and reach real model generalizability and fairness. Self-supervision may be one of the cornerstones for reaching this goal. Second, this study is the first of its kind to use a dataset of more than 30K BWRs and labels that are based on EHR. EHR data is fairly abundant in dentistry, yet EHR-based labels are associated with uncertainties and biases [36]. We in parts circumvented the lack of diagnostic consistency by establishing a test set of 343 BWRs with a more sophisticated ground truth. On the other hand, as we used EHR-based label for training of the models a degradation of the model performance could be expected. This may be the reason why our results are slight worse than, e.g., presented by Lee et al. [56]. They trained a classification model for caries on tooth segments from 3000 periapical radiographs and reported values of 81.0% (95% CI 74.5–86.1%), 83.0% (95% CI 76.5–88.1%), and 84.5% (95% CI 79–90%) for sensitivity, specificity, and ROC-AUC. It is noteworthy that they used a clean and prepossessed dataset with strong exclusion criteria, which we did not. It remains to be seen which metrics can be achieved once AI-based systems enter clinical practice and are evaluated on prospectively collected data. One way to improve our approach would be to validate and re-label (e.g., following the same labelling procedure as outlined in this study) larger parts of the dataset and fine-tune the model with reviewed labels. Third, the present study evaluated aspects of data-efficiency and the value of data for commonly used dataset sizes (up to 3.8K BWRs). Such results are relevant for planning future data-collection and data-labeling initiatives as, no matter how hard the community will try, unlabeled data will always outnumber labeled data by far. Lastly, this study emphasized the need for reporting prevalence in the datasets that are used for training machine learning models. Researchers should report robust metrics, such as F1-score or ROC-AUC next to metrics, such as sensitivity and specificity, among others. For low-prevalence classification tasks, commonly applied strategies such as random data selection may not be the most promising ones. Balancing the positive and negative cases in the training set should be explored, although, we emphasize that the test set should never be oversampled but should correspond to the target population.

Next to these strengths, the study comes with a few weaknesses. First, the usage of EHR labels is problematic, as they stem from routine care and are affected by biases, incompleteness, inconsistencies, and limited accuracy. However, on the other side, this is “real” data as it is stored in large amounts in data silos all over the world. It would be a pity if we as a community could not make use of this treasure trove of data. Second, we developed models for caries classification on the tooth level. As a downside, the first one needs to have an understanding of the tooth as an object on the BWR. In this study we used a helper model to crop tooth segments from BWRs, a work that could not have been done manually with ease due to the large amount of data. Further, in a clinical setting, tooth-wise caries classification may be of limited value as a more detailed assessment, e.g., on the tooth surface level, would be expected by the practitioner. Lastly, we did not apply any attempts to investigate our outcomes with techniques of Explainable AI. Such techniques help to find biases and yield insight into the decision-making processes of the model [57].

## 5. Conclusions

This work explored whether recent advancements in self-supervised learning algorithms could benefit the performance and label efficiency in dental caries classification scenarios. We demonstrated the obtained gains by fine-tuning (transfer learning) the obtained data representations to dental caries classification. Our evaluation of multiple hypotheses showed that pretraining models with self-supervised methods using unlabeled data outperforms non-pretrained counterparts. What is more, our results, particularly in the annotation-efficient regime, demonstrated the possibility to reduce the manual annotation effort required in the medical imaging domain, where annotation scarcity is an obstacle. Moreover, our results showed that even noisy EHR-based labels are sufficient for transfer learning with self-supervised models for dental caries classification, further reducing the need for human expert annotations. Our work presented a framework for exploiting inexpensive unlabeled data and also noisy EHR labels, and is a first step toward utilizing self-supervised learning methods for label-efficient clinically-relevant applications.

## Figures and Tables

**Figure 1 diagnostics-12-01237-f001:**
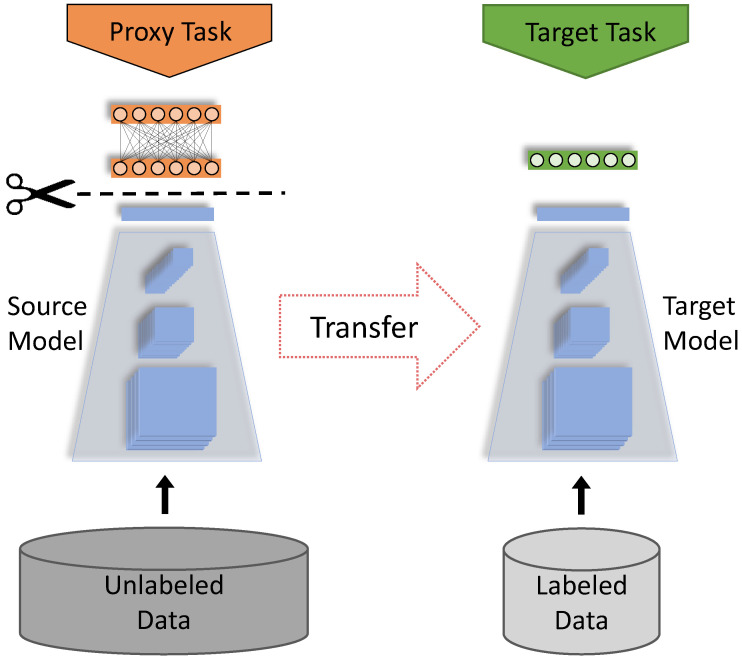
Flowchart of self-supervised learning stages. First, a deep learning model, e.g., CNN, is trained on unlabeled data using a proxy task. Then, the obtained knowledge (representations) is transferred into a target downstream task.

**Figure 2 diagnostics-12-01237-f002:**
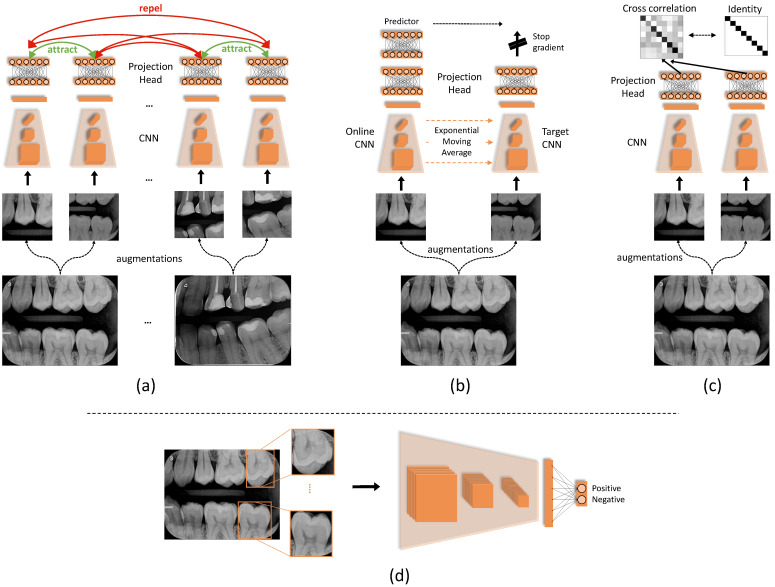
Illustration scheme of the three self-supervised algorithms and how to fine-tune the resulting encoder CNN. (**a**) SimCLR relies on attracting the views of each image together and repelling them from the views of other images. (**b**) In BYOL the target network calculates moving averages of the online network, which is updated with loss gradients. (**c**) Barlow Twins computes the cross-correlation matrix of two batches of image views and minimizes its difference to the identity matrix. (**d**) The obtained CNN encoder is fine-tuned on input tooth images for caries classification.

**Figure 3 diagnostics-12-01237-f003:**
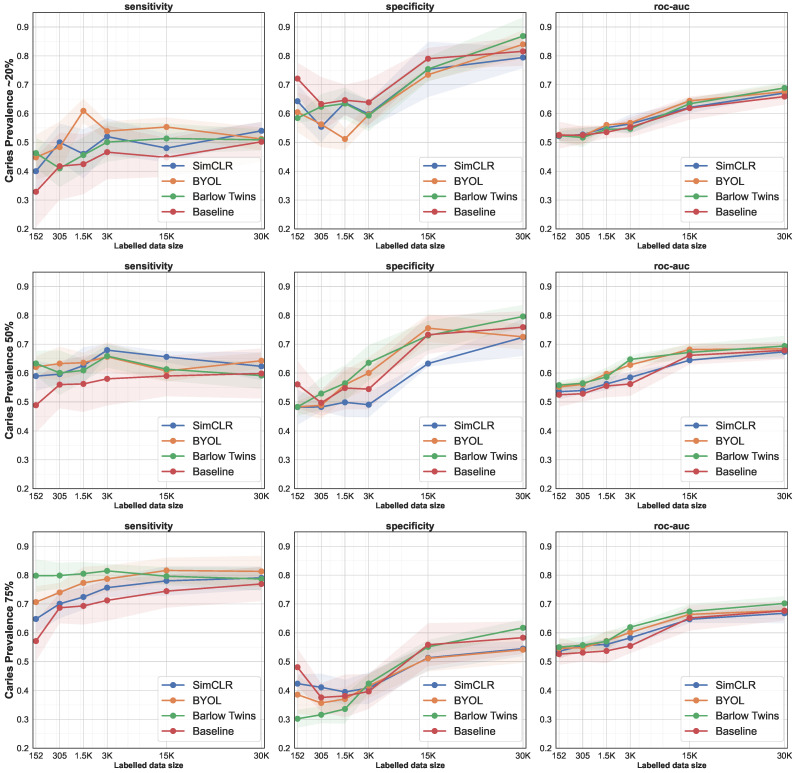
Evaluation results for data-efficiency by successively increasing the size of the training set. Each row represents a different caries prevalence group (∼20%, 50%, 75%), and the columns are evaluation metrics (sensitivity, specificity, roc-auc). The plots show the mean for each metric in thick lines and the 95% confidence interval (CI) as bands.

**Table 1 diagnostics-12-01237-t001:** Caries classification results when fine-tuning on the full training set. We highlight in **bold** the best models.

Method	Sensitivity	Specificity	ROC-AUC
Baseline	51.80	**91.30**	71.50
SimCLR	57.20	89.30	73.30
BYOL	54.60	**91.30**	73.00
Barlow Twins	**57.90**	88.90	**73.40**

**Table 2 diagnostics-12-01237-t002:** Caries classification results when fine-tuning on varying quantities of labeled samples (numbers of #teeth/#BWRs). The results are grouped by the prevalence of the caries (∼20%, 50%, 75%). We highlight in **bold** the best models in each row (i.e., for each fine-tuning dataset size).

Prev.	#Teeth/#BWRs	SimCLR			BYOL			Barlow Twins			Baseline		
		Sens.	Spc.	Roc	Sens.	Spc.	Roc	Sens.	Spc.	Roc	Sens.	Spc.	Roc
∼20%	152/18	40.02	64.27	52.15	44.78	60.45	**52.61**	**46.28**	58.43	52.35	32.87	**72.10**	52.49
	305/37	**50.05**	55.39	**52.72**	48.35	56.26	52.31	41.01	62.34	51.68	41.74	**63.30**	52.52
	1.5K/190	46.40	63.78	55.09	**60.92**	51.16	**56.04**	45.60	63.46	54.53	42.45	**64.62**	53.53
	3K/380	52.99	59.79	56.39	**53.88**	59.61	**56.75**	50.08	59.29	54.69	46.61	**63.86**	55.23
	15K/1.9K	48.96	75.28	62.12	**55.32**	73.44	**64.38**	51.34	75.40	63.37	44.78	**79.00**	61.89
	30K/3.8K	**54.80**	79.42	67.11	51.18	83.98	67.58	50.88	**86.85**	**68.87**	50.19	81.57	65.88
50%	152/18	58.94	48.19	53.56	62.09	48.28	55.19	**63.34**	48.28	**55.81**	48.85	**56.09**	52.47
	305/37	59.62	48.24	53.93	**63.29**	48.83	56.06	60.07	**52.89**	**56.48**	56.00	49.69	52.84
	1.5K/190	62.59	49.85	56.22	**63.58**	55.93	**59.75**	60.92	**56.44**	58.68	56.24	54.80	55.52
	3K/380	**67.95**	49.03	58.49	65.65	60.02	62.83	65.91	**63.56**	**64.73**	58.00	54.43	56.21
	15K/1.9K	**65.62**	63.29	64.46	60.71	**75.57**	**68.14**	61.34	73.01	67.17	58.99	73.28	66.13
	30K/3.8K	62.33	72.40	67.37	**64.28**	72.57	68.42	59.13	**79.63**	**69.38**	59.86	75.90	67.88
75%	152/18	64.80	42.37	53.59	70.64	38.56	54.60	**79.81**	30.20	**55.01**	57.15	**48.07**	52.61
	305/37	70.07	**41.10**	55.59	74.00	35.67	54.84	**79.86**	31.59	**55.73**	68.68	37.58	53.13
	1.5K/190	72.42	**39.49**	55.96	77.32	37.04	57.18	**80.49**	33.59	**57.04**	69.32	38.10	53.71
	3K/380	75.65	40.85	58.25	78.68	41.59	60.14	**81.48**	**42.41**	**61.95**	71.25	39.66	55.45
	15K/1.9K	78.02	51.35	64.69	**81.62**	51.16	66.39	79.62	55.13	**67.38**	74.45	**55.86**	65.15
	30K/3.8K	79.04	54.51	66.77	**81.29**	54.14	67.72	78.66	**61.74**	**70.20**	76.94	58.31	67.62

## Data Availability

The data are not publicly available.

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
