# Peer review of "Self-Supervised Learning Methods for Label-Efficient Dental Caries Classification"

_diagnostics, 2022, doi:10.3390/diagnostics12051237_

Round 1

Reviewer 1 Report

The article complies with the journal requirements and can be published.

The lack of data annotations is one of the major hurdles for machine learning applications in healthcare. At the same time, a huge amount of unlabeled data accumulates in clinical procedures, which requires solutions to use a lot of data. The way out is to use self-learning algorithms to label data. This is the relevance and novelty of this work. 
The results of the work demonstrated the possibility of reducing the manual annotation required in the field of medical imaging, where the lack of annotations is a major obstacle. This is the reason for the interest in this work of many health professionals.

Author Response

We thank the reviewer for their comments. We are encouraged, as they also see the value in our work and they find it interesting for the field. Self-supervised learning offers the means to bridge the gap between the large quantities of unlabeled data and the relatively few annotated samples. 

Reviewer 2 Report

This paper studies “Self-Supervised Learning Methods for Label-Efficient Dental Caries Classification”- This study shows that self-supervision can provide gains in medical image 14 analysis, particularly when obtaining labels is costly and expensive. It is an interesting topic. However, I suggest the authors resubmit it after a minor revision. My suggestions are as follows:

1. The paper should be revised to include at least 10 recent references (2021-2022) based on the suggested topic.

2. I strongly suggest that the paper be proofread and reread meticulously again, particularly regarding the spelling and grammatical mistakes.

3. Your introduction is too short. Please add at least two paragraphs at the end of the introduction and explain the methodology.

4. Please explain the structure of the paper at the end of the introduction.

5. In some cases, such as Figure 1, the reference to figures is forgotten.

6. In lines 131-132 you mentioned “Therefore, BYOL introduces asymmetric parameter updates to the encoder architecture as an alternative.”

Please explain and clarify more. What is your mean by mentioning “encoder architecture as an alternative”? this part needs more clarification and explanation.

7. Your topic for Fig 1. And Fig 2. is too long. Please convert this explanation to your text.

8. Flowchart is beneficial, it’s also important to outline the methodology behind the algorithm. Please provide a flowchart after the introduction for more clarification.

9. When you mentioned “attracting the views (augmented versions)” in Figure 1, what exactly do you mean? Please explain more.

Author Response

We thank the reviewer for their insightful feedback. We are encouraged, as they find the topic of our work interesting and relevant for the domain. Regarding their comments to improve the manuscript, we have addressed them now in the revised version. We also left some comments in the pdf to highlight which comment each change addresses. In addition, we added a flowchart, as suggested by the reviewer. Finally, we should mention that, regarding point #5, that there we reference Figure 1 in the manuscript, inside each algorithm's section. Note that this figure is called Figure 2 now, due to adding the flowchart in the introduction section.